# Two-Path 77-GHz PA in 28-nm FD-SOI CMOS for Automotive Radar Applications

**Claudio Nocera [1], Giuseppe Papotto [1] and Giuseppe Palmisano [2],***

[1] STMicroelectronics, 95121 Catania, Italy; claudio.nocera@st.com (C.N.); giuseppe.papotto@st.com (G.P.)
[2] Dipartimento di Ingegneria Elettrica Elettronica e Informatica (DIEEI), University of Catania, 95125 Catania, Italy
* Correspondence: giuseppe.palmisano@unict.it

**Abstract:** This paper presents a 77 GHz two path power amplifier (PA) for automotive radar applications. It was fabricated in 28-nm fully depleted silicon-on-insulator CMOS technology, which provides transistors with a transition frequency of about 270 GHz and a general-purpose low cost back-end-of-line. The proposed PA consists of a 50 Ω input buffer followed by two power units, which are made up of a current-reuse common source driver for improved efficiency and a stacked cascode power stage for enhanced output power. A peak detector was also embedded into the PA for output power monitoring. The designed PA achieved a saturated output power as high as 17.4 dBm at 77 GHz with an excellent power added efficiency of 19%, while drawing 150 mA from a 2 V power supply. The core die size was 500 μm × 300 μm.

**Keywords:** two path mm-wave PA; automotive radar sensor; CMOS technology; current combining PA; electromagnetic (EM) simulations; mm-wave circuit

## 1. Introduction

RADAR sensors are key devices for the development of next generation advanced driving assistance systems (ADAS). The main goal of such systems is to provide fully autonomous driving, which may potentially reduce car accidents, thus preventing injuries and saving human lives [1]. ADAS relies on a network of several different sensors (i.e., lidar, radar, video, infrared, etc.) distributed around the car, which collect data from the surroundings of the vehicle, thus enabling a microcontroller to properly drive its motion. Radar devices are crucial for the implementation of reliable self-driving systems. Indeed, they guarantee high robustness over environmental interferences (i.e., poor light, extreme temperature, bad weather conditions, etc.), and hence they can operate where other technologies would fail. A common ADAS implementation involves radar sensors with different requirements on resolution and operating distance. To meet these requirements while constraining the system cost, multimode radar solutions are preferred. They can support both long-range radar (LRR, from 76 to 77 GHz) and short-range radar (SRR, from 77 to 81 GHz) operation modes, thus avoiding the need for different radar devices.

Recently, fully integrated and energy efficient CMOS radar sensors operating at W-band have been demonstrated [2–5]. However, the implementation of self-driving systems poses severe design challenges. The main issue is related to the extension of the radar operating range, namely the maximum distance at which a target can be detected. An effective autonomous driving system calls for radar sensors able to guarantee an operating range as high as 250 m. This translates into stringent requirements for the RX sensitivity, $S_{RX}$, and mainly for the TX output power, $P_T$. Specifically, in typical application scenarios, assuming an $S_{RX}$ of −110 dBm, the TX should be able to provide a $P_T$ as high as 13 dBm to enable an operating distance of 250 m [6]. Unfortunately, the TX power is affected by the package insertion loss besides suffering from process, supply voltage, and temperature (*PVT*) variations. As a result, the power constraint for the PA of a modern radar sensor

turns out to be much more challenging. Indeed, although radar sensors usually exploit advanced flip-chip assembly techniques, a package insertion loss of around 1 dB is to be taken into account in the PA power budget. Moreover, the PA output power exhibits an additional reduction up to 3 dB, which is due to *PVT* variations [7]. As a result, the PA should be able to deliver an output power higher than 17 dBm to comply with the targeted 250 m operating range. Unfortunately, this requirement cannot be achieved with single path CMOS PAs in a low voltage scaled CMOS implementation [6,8,9].

The PA output power is set by the size of the output transistors and the equivalent load resistance provided by the inductive components (i.e., inductors and transformers) in the resonant load. Unfortunately, the larger the transistor size, the greater the load capacitance, and this leads to a reduction of the load inductance and hence of the equivalent load resistance. This ultimately limits the drain voltage swing of the power transistors and hence the output power. To overcome such a limitation, power combined multi path PAs have been proposed [10,11]. By properly combining two or more power stage units, they allow the output power to be boosted above the maximum achievable value by a single path PA.

In this work, a two path 77 GHz PA is presented, which consists of a 50 Ω input buffer driving two power units. The latter are made up of a common source (CS) driver stage with current reuse for improved efficiency and a stacked cascode (CAS) power stage for enhanced output power. A peak detector was also included, which allows power monitoring. The circuit adopts a 2 V supply voltage and was fabricated in a 28-nm fully depleted silicon-on-insulator (FD-SOI) CMOS technology with low cost back-end-of-line (BEOL).

The paper is organized as follows. Section 2 presents the adopted power stage along with the best state-of-the-art solutions. The proposed PA is presented in Section 3, where the design of passive components is also discussed. Experimental results and comparison with the state of the art of 77 GHz PAs are given in Section 4. Finally, conclusions are drawn in Section 5.

## 2. Power Stage Architecture

The power stage is the key circuit of a PA, since it mainly contributes to the performance of both power level and efficiency. Therefore, it calls for a careful design especially in mm-wave applications. Typical power stage topologies at W-band are shown in Figure 1, along with the proposed one.

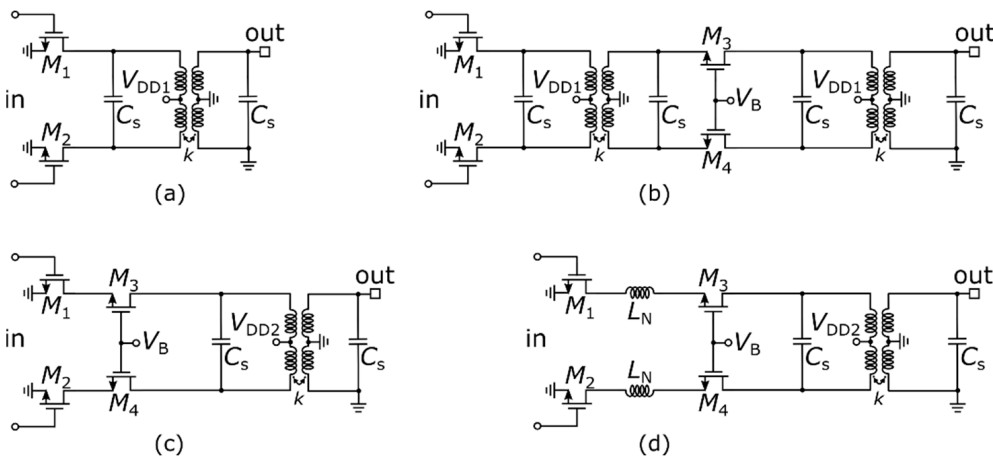

**Figure 1.** Mm-wave power stage topologies. (**a**) CS stage; (**b**) TBFC stage; (**c**) CAS stage; (**d**) enhanced CAS stage.

Whatever the adopted solution, the optimization of the PA performance in nano meter CMOS technologies needs a proper biasing of the power stage transistors, which has to provide optimum current density and maximum drain-source voltage [12]. Indeed, such

a bias condition enhances the transistor $f_T$, thus improving gain and power performance. Consequently, the traditional CS and the transformer-based folded-cascode (TBFC) [6] topologies in Figure 1a,b, respectively, prove to be especially suitable for W-band power amplifications with low voltage implementations. Indeed, they guarantee optimum bias conditions for the power transistors with the minimum supply voltage, $V_{DD1}$. However, the CS topology suffers from pour reverse isolation that leads to stability issues. Conversely, the common source stage in the TBFC configuration develops the highest transconductance gain while avoiding high voltage gain. This allows maximizing the output current of the input pair, $M_{1,2}$, without stability issues. However, the TBFC topology relies on an inter-stage transformer-based matching network, whose insertion loss affects the gain performance, thus limiting the maximum output current. Stacking transistors as in the cascode stage (CAS) architecture in Figure 1c allows for overcoming this limitation at the cost of a higher power supply with respect to the TBFC topology. Indeed, to guarantee optimum biasing for the power transistors in a CAS topology a supply voltage, $V_{DD2}$, higher than $V_{DD1}$ should be used. However, since both common source and common gate stages in the CAS architecture share the same quiescent current, TBFC and CAS topologies exhibit almost the same power consumption.

An enhanced CAS topology was adopted for the power stage of the proposed PA, which consists of a stacked cascode topology with inter-stage series inductors, $L_N$, as shown in Figure 1d. Actually, the conventional CAS architecture suffers from the low impedance paths towards ground caused by the parasitic capacitances in the drain nodes of the input pair, $M_{1,2}$. These paths significantly impact the power stage performance at mm-wave frequencies. To overcome such a limitation, series inductors $L_N$ were used between the common source and common gate stages to compensate for the overall parasitic capacitance in the inter-stage node. Moreover, inductors $L_N$ perform an upward transformation of the real part of the input impedance of the common gate transistors, $M_{3,4}$, thus improving power transfer. A supply voltage of 2 V was used in this design, which is twice the 1 V voltage headroom allowed by the technology. However, optimum biasing and safe operating conditions for transistors $M_{1-4}$ were guaranteed by properly setting the bias voltage, $V_B$.

## 3. Circuit Description

The proposed two path PA was designed using a 28-nm FD-SOI CMOS technology by STMicroelectronics, which provides 1 V low-$V_t$ transistors and a low-cost general-purpose BEOL [13]. The lack of dedicated thick copper metal layers, typically available in custom mm-wave technology platforms, makes passive components and layout design crucial. Moreover, layout symmetry plays an important role in the design of effective power-combined PAs. Specifically, combining PA power units demands a high layout symmetry and a complex routing to preserve in-phase signals and optimize circuit performance. To this aim, extensive EM and post layout simulations were carried out.

### 3.1. Power Amplifier Design

The schematic of the proposed two path PA is shown in Figure 2. It consists of a 50 Ω input buffer that drives two PA units arranged in a current-combining configuration to boost the power delivered to the output load. Each PA unit comprises a driver and a power stage. The input buffer exploits a cascode topology with the input balun, $T_{IN}$, that provides both single-to-differential ended conversion and ESD protection, and inductor $L_s$ that sets the real part of the input impedance to 50 Ω. Capacitor $C_S$ between the gate terminals of $M_{1,2}$ compensates for the imaginary part of the input impedance.

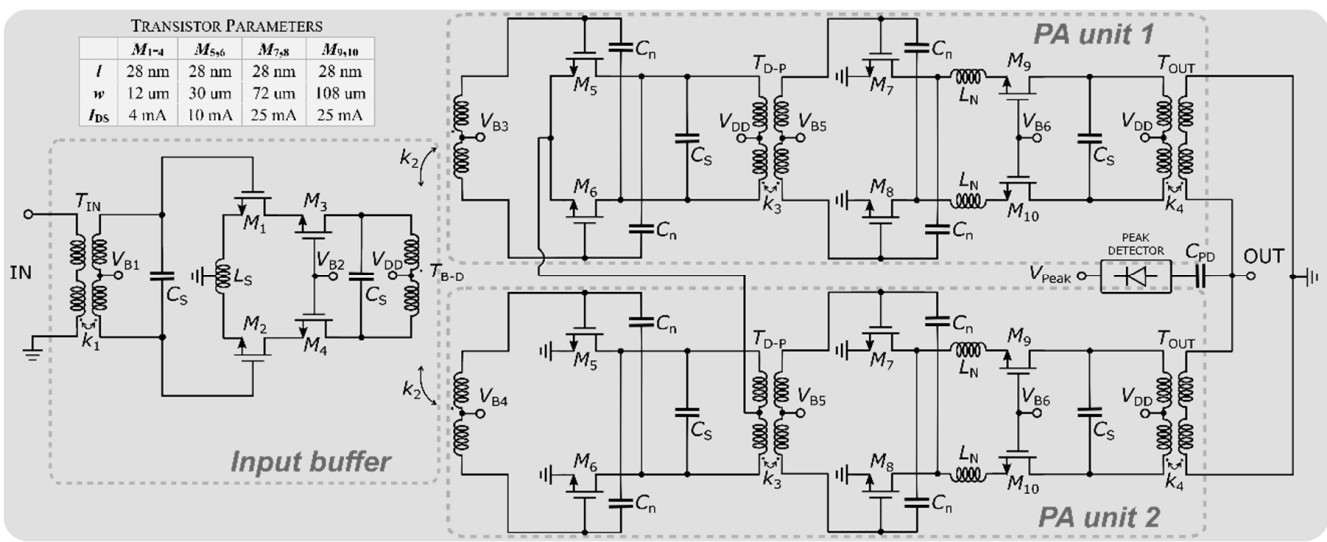

**Figure 2.** Schematic of the proposed two path PA.

The buffer is loaded by a three-way transformer, $T_{B–D}$, which allows the two PA units to be driven by identical input signals. Specifically, the primary winding of $T_{B–D}$ provides the input buffer with a proper resonant load, whereas the secondary windings share equally the buffer output signal to the PA units. The driver stages in the PA units adopt a CS topology and exploit a transformer-coupled current-reuse approach to improve the PA efficiency [14]. Specifically, the central tap of the primary coil of the load transformer, $T_{D–P}$, in the driver stage of PA unit 2 is shorted to the source node of the driver stage of PA unit 1. The former is then supplied by $V_{DD}$ through the central tap of the primary coil of its load transformer, $T_{D–P}$. As a result, the two driver stages share the same quiescent current, thus saving power and improving efficiency. Neutralization capacitors, $C_n$, were used to compensate for gate-drain capacitances of transistors $M_{5,6}$. This results in a higher input/output isolation, which improves frequency stability and gain performance. Moreover, the neutralization technique prevents the driver equivalent input capacitance from being increased by the Miller effect, thus leading to a more robust inter-stage matching between buffer and driver stages [15]. The power stage of each PA unit adopts the enhanced CAS topology in Figure 1d, in which neutralization capacitors were also included. Bias voltage $V_{B6}$ was set to provide transistors $M_{7–10}$ with a drain-source bias voltage close to 1 V to guarantee optimum operation as mentioned before, while providing safe static conditions. Indeed, this allows simultaneously achieving improved frequency performance for transistors $M_{7,8}$ and best output voltage swing for transistors $M_{9,10}$. The handling current capability of the power stage, and ultimately of the PA, is set by the aspect ratios of transistors $M_{7,8}$ and $M_{9,10}$. Unfortunately, transistors $M_{7,8}$ and $M_{9,10}$ give the dominant contribution to the capacitances at the input and inter-stage matching networks of the driver, respectively. Therefore, larger power transistors translate into larger capacitances to be driven. Actually, a proper value exists for the aspect ratios of transistors $M_{7,8}$ and $M_{9,10}$, which optimizes transconductance gain, thus achieving maximum output power. The transistors parameters of the PA stages are reported in Figure 2. Minimum channel length and optimum current density were used for all the PA transistors to achieve the maximum $f_T$.

The two PA units are coupled to the external load through transformers $T_{OUT}$ that perform a current-combining structure, which simultaneously provides impedance matching, power combination, biasing and differential-to-single ended conversion. Finally, a peak detector was also embedded at the PA output for power monitoring, whose simplified schematic is shown in Figure 3. It exploits a feedback solution that improves accuracy over temperature and process variations. Specifically, the common-drain transistor, $M_1$, along with capacitor $C_3$, performs the peak detection of the PA output voltage at its source

node. The rectified peak voltage is accurately replicated at the source node of $M_2$ and then delivered to the output, thanks to the local feedback loop based on an error amplifier (EA).

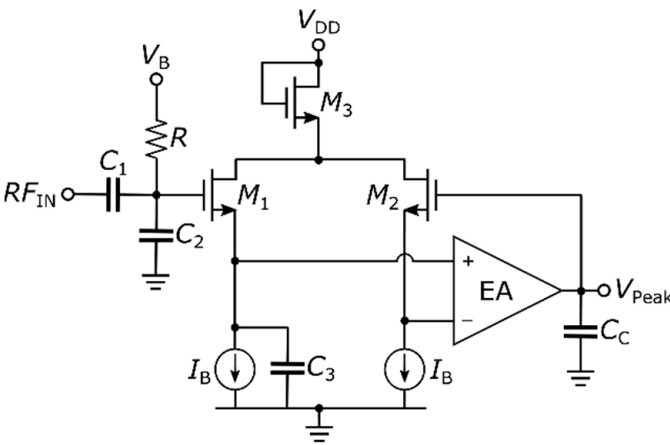

**Figure 3.** Simplified schematic of the peak detector.

The peak detector provides a sensed amplitude that is compensated with respect to process and temperature variations. Actually, such variations appear as a common-mode signal at the input of the error amplifier and hence they are rejected. Thin oxide minimum channel length devices were used for $M_{1,2}$ to minimize the capacitive contribution of the peak detector to the PA output matching network. To the same purpose, a coupling capacitance, $C_1$, as low as 10 fF was used, which performs with capacitor $C_2$ a $1/3$ capacitive partition. This properly limits the input signal to transistor $M_1$, thus preventing undesired gate oxide breakdown. Finally, diode connected transistor $M_3$ was stacked over $M_{1,2}$ to allow the peck detector to share the same 2 V supply voltage of the PA.

### 3.2. Mm-Wave Transformer Design

The performance of mm-wave circuits is strongly affected by the quality of passive components, especially inductors and transformers. Moreover, in the case of multi path PAs a symmetrical layout is also crucial to preserve in-phase signals for an effective power-combining operation. In this work, multilayer inductive components were designed, which exhibit high symmetry while providing both high quality factor (*Q*) and self-resonant frequency (*SRF*). Inductors and transformers exploit the last three metal layers at the top of the stack of the adopted CMOS technology, with the aim of reducing the series resistive losses at the cost of a slightly lower *SRF*. No patterned ground shield (PGS) was used since it causes detrimental effects at mm-wave [16]. Figure 4a shows the layout view of the proposed power splitter transformer, $T_{\text{B–D}}$. Basically, it consists of a three-way stacked transformer with the secondary windings that are interleaved with each other and stacked to the primary winding. Thanks to this arrangement, an effective and compact size power splitter was attained.

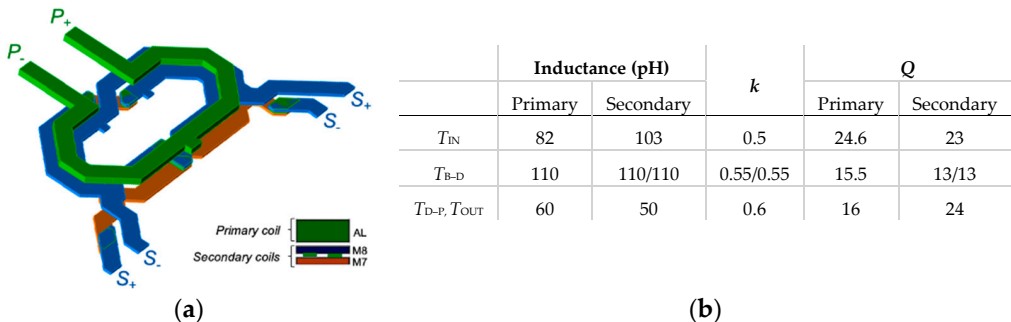

| | Inductance (pH) | | *k* | *Q* | |
|---|---|---|---|---|---|
| | Primary | Secondary | | Primary | Secondary |
| $T_{\text{IN}}$ | 82 | 103 | 0.5 | 24.6 | 23 |
| $T_{\text{B–D}}$ | 110 | 110/110 | 0.55/0.55 | 15.5 | 13/13 |
| $T_{\text{D–P}}, T_{\text{OUT}}$ | 60 | 50 | 0.6 | 16 | 24 |

(**a**)  (**b**)

**Figure 4.** (**a**) Power splitter transformer, $T_{\text{B–D}}$. (**b**) Electrical parameters of transformers at 77 GHz.

An interleaved structure was preferred for transformer $T_{\text{IN}}$ to achieve higher $Q$ for both primary and secondary coils, which is useful for 50 Ω input matching. Conversely, a stacked configuration was adopted for transformers $T_{\text{D–P}}$ and $T_{\text{OUT}}$, since it provides higher coupling factor, $k$, thus guaranteeing maximum power transfer and better area occupancy with respect to interleaved structures. The coil width and inner diameter of each transformer were set to achieve both maximum $Q$ at 77 GHz and a *SRF* higher than twice the operating frequency. The simulated electrical parameters of the transformers designed for this work are summarized in Figure 4b.

Finally, Figure 5a displays the 3D layout view of the passive circuitry of the two power stages, which includes inductors, transformers, and interconnections among components. This layout was used for EM simulations that were carried out to accurately account for both layout parasitics and undesired coupling effects. An *S*-parameter model was extracted from these simulations and embedded into the schematic entry to tune transistor aspect ratios and matching networks for the optimum PA performance.

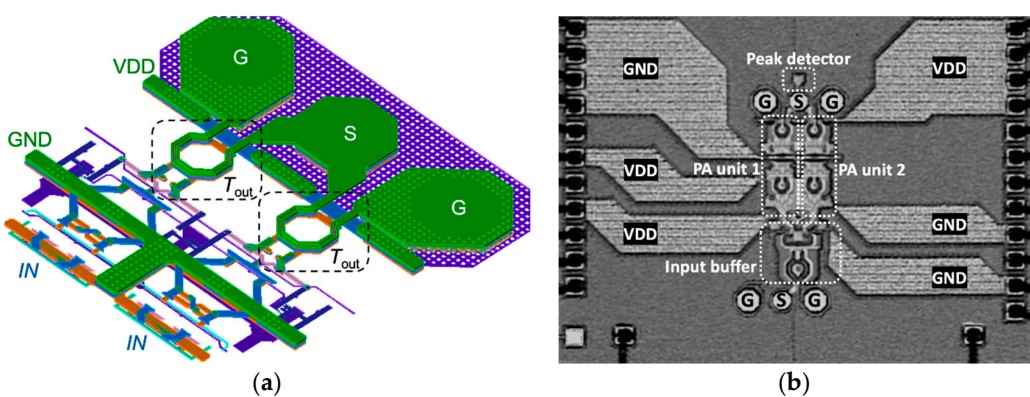

(**a**)　　　　　　　　　　　　　　　　　　　　　　(**b**)

**Figure 5.** (**a**) 3D view of the passive circuitry of the power stages. (**b**) Die photograph of the 77 GHz PA.

## 4. Experimental Results

The proposed two path PA was fabricated in 28-nm FD-SOI CMOS technology by STMicroelectronics, featuring a general-purpose, low cost BEOL with eight copper metal layers plus an aluminium top one, and metal-oxide-metal (MOM) capacitors with a 5 fF/μm$^2$ of specific capacitance. The microphotograph of the designed PA is shown in Figure 5b. The die area is pad limited for testing purposes, whereas the amplifier core size is 500 μm × 300 μm. The chip was mounted on a FR4 printed circuit board (PCB) and all the input/output dc pad were wire bonded. Conversely, the input/output mm-wave signals were directly probed by using a Cascade probe station and dedicated GSG on-chip pads.

The PA draws a quiescent current of 150 mA from the 2 V power supply. Figure 6a shows the measured *S*-parameters. As is apparent, good input and output 50 Ω matching is provided. Actually, the measured input ($-S_{11}$) and output ($-S_{22}$) return losses at 77 GHz are around 30 dB and 11 dB, respectively. A small signal gain as high as 28 dB is achieved with a reverse isolation ($-S_{12}$) that is higher than 40 dB. The PA output power as a function of the frequency is displayed in Figure 6b. The output power curve is almost flat over a wide frequency range. Specifically, it exhibits a variation lower than 0.5 dB in the band of interest, ranging from 76 GHz to 81 GHz.

Figure 7a shows the measured output power and power added efficiency (PAE) as a function of the input power. The PA is able to deliver a maximum saturated output power, $P_{\text{sat}}$, of 17.4 dBm with an excellent PAE of 19%. The peak detector output voltage as a function of the output power is plotted in Figure 7b.

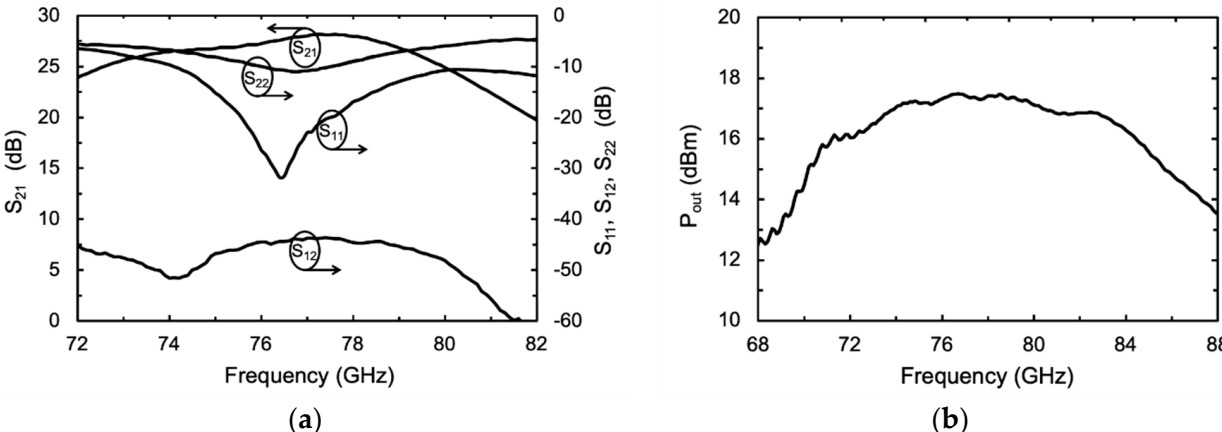

**Figure 6.** (**a**) Measured S-parameters of the 77 GHz PA. (**b**) Measured PA output power versus frequency.

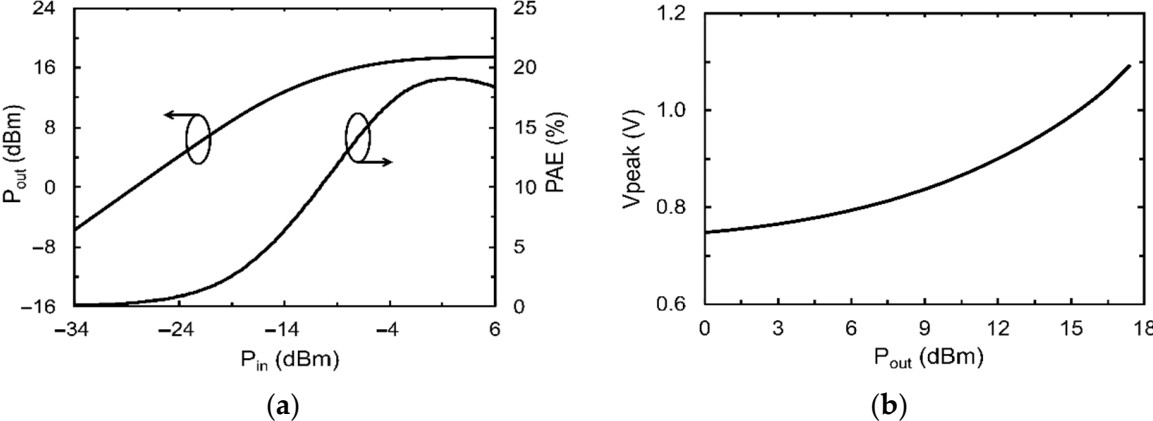

**Figure 7.** (**a**) Measured PA output power and PAE versus input power. (**b**) Measured peak detector voltage versus output power.

Reliability tests were carried out. Specifically, the PA power supply was increased up to 2.5 V without any failure. For the sake of completeness, $P_{sat}$ as a function of the supply voltage is plotted in Figure 8a. Moreover, the values of $P_{sat}$ and PAE as a function of temperature are shown in Figure 8b.

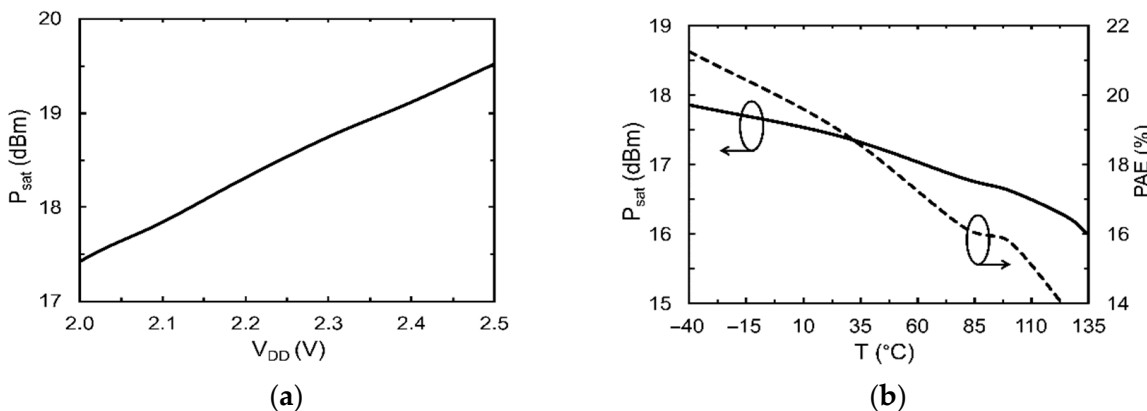

**Figure 8.** (**a**) Measured $P_{sat}$ versus supply voltage. (**b**) Measured $P_{sat}$ and PAE versus temperature.

Table 1 summarizes the PA experimental results while comparing them with the state-of-the-art 77 GHz CMOS PAs. The proposed PA exhibits the best-in-class performance in terms of $P_{sat}$, PAE and linear gain. A higher $P_{sat}$ is achieved by [17] thanks to the 4-path architecture. However, the output power of multi path PAs as well-known increases of

around 2.5 dB for each couple of paths. This suggests that the normalized output power, $P_N$, over a 2-path configuration has to be considered for a fair comparison. As a result, the work in [17] achieves about the same $P_N$ as the proposed PA but with much lower efficiency.

**Table 1.** Performance comparison with the state-of-the art 77 GHz PAs.

| Ref. | Tech. | Amplifier Topology | Power Combining | Peak Detector | Freq [GHz] | $P_{sat}$ [dBm] | $P_N$ [dBm] | G [dB] | PAE [%] | $V_{DD}$ [V] | $P_{dc}$ [W] | Core Die Size [mm²] | FoM * |
|---|---|---|---|---|---|---|---|---|---|---|---|---|---|
| [17] | 40-nm CMOS | Doherty with Cascode amplifier | 4-path | n.a | 77 | 20 | 17.5 | 20 | 12 | 1.5 | 0.8 | 0.19 | 7115 |
| [18] | 65-nm CMOS | 2-Cascode | 2-path | n.a | 77 | 12 | 12 | 25 | 4.2 | 1.8 | 0.35 | 0.34 | 1248 |
| [19] | 90-nm CMOS | 1-CS+ 1-Cascode | 4-path | n.a | 77 | 12.2 | 9.7 | 24.1 | 17.3 | 2.4 | 0.09 | 0.73 | 4375 |
| [20] | 55-nm CMOS | 2-Cascode | 2-path | n.a | 77 | 15 | 15 | 15 | 8 | 2.5 | n.a | 0.21 | 474 |
| [2] | 65-nm CMOS | 4-CS | 2-path | n.a | 77 | 15.4 | 15.4 | 24.4 | 10.4 | 1 | 0.33 | - | 5889 |
| [21] | 65-nm CMOS | 2-Cascode | 2-path | n.a | 77 | 15.8 | 15.8 | 21 | 15.2 | 2 | 0.25 | 0.21 | 4313 |
| This Work | 28-nm CMOS | 1-Cascode+ 1-CS +1-Cascode | 2-path | a | 77 | 17.4 | 17.4 | 28 | 19 | 2 | 0.3 | 0.15 | 39,060 |

* FoM $= P_{sat} \times G \times \text{PAE} \times f^2$ [22].

## 5. Conclusions

This paper presents a two path 77 GHz PA implemented in 28 nm FD-SOI CMOS technology. The PA is based on a two-path current-combining approach and uses an input buffer driving two power units. The latter are made up of a common source driver stage with current reuse and a power stage exploiting an enhanced CAS topology. Despite the general-purpose BEOL of the adopted technology, the proposed PA exhibits best-in-class performance in terms of both saturated output power and efficiency. Indeed, it exhibits a power gain of 28 dB and is able to deliver a saturated output power of 17.4 dBm with an excellent PAE of 19%.

**Author Contributions:** Conceptualization, G.P. (Giuseppe Papotto); methodology, C.N. and G.P. (Giuseppe Papotto); validation, C.N.; formal analysis, C.N. and G.P. (Giuseppe Papotto); investigation, C.N.; writing—original draft preparation, C.N., and G.P. (Giuseppe Papotto); writing—review and editing, C.N. and G.P. (Giuseppe Palmisano); supervision, G.P. (Giuseppe Palmisano); project administration, G.P. (Giuseppe Palmisano). All authors have read and agreed to the published version of the manuscript.

**Funding:** This research received no external funding.

**Acknowledgments:** The authors would like to thank M. Rizzo, STMicroelectronics, Catania, for layout support. A special thank you also to A. Castorina and A. Michelin, STMicroelectronics, Catania, for measurement assistance.

**Conflicts of Interest:** The authors declare no conflict of interest.

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
