# Peer review of "Two-Path 77-GHz PA in 28-nm FD-SOI CMOS for Automotive Radar Applications"

_electronics, doi:10.3390/electronics11081289_

Round 1

Reviewer 1 Report

How about die size comparison with the other published articles at Table I?

Author Response

How about die size comparison with the other published articles at Table I?

  • Thanks for your comment. For the sake of completeness, the die size comparison has been included in Table I

Please check the attached file for the replies. 

Reviewer 2 Report

In row 41, the term "PVT" was not defined previously.

Please show more in-depth the design procedure:

  1. Show how stable your circuit is and how the capacitor Cn helps to compensate your circuit and what size it should be.

  1. Since this circuit is operating with a supply voltage higher than the maximum safe supply voltage for this CMOS process. Show how this circuit should be biased (bias voltage calculation).

  1. The purpose of this circuit is automotive applications. The automotive industry is one of the most demanding industries and the circuits used in this field must show a high degree of stability and safe operation. How can your proposal meet the safe operation requirements of this industry and overcome the concern associated with the correct and stable biasing?

  1. How process variations can affect your circuit performance and the risk associated with correct circuit biasing.

Author Response

In row 41, the term "PVT" was not defined previously.
• PVT has been defined (lines 45-46) in the revised manuscript.
Please show more in-depth the design procedure:
1. Show how stable your circuit is and how the capacitor Cn helps to compensate your circuit and what size it should be.
• The use of neutralization capacitors, Cn, to compensate for gate–drain capacitances of a transistor pair is a well-known stabilization technique. Indeed, it improves isolation between input and output terminals, thus guarantying better frequency stability, as stated on page 4 from line 146 to 151. Anyway, a new reference [15] has been included for better clarity.
2. Since this circuit is operating with a supply voltage higher than the maximum safe supply voltage for this CMOS process. Show how this circuit should be biased (bias voltage calculation).
• A supply voltage of 2 V was used in this design, which is twice the 1-V voltage headroom allowed by the technology under static conditions. However, optimum biasing and safe operating conditions for transistors were guaranteed by properly setting the CG transistor bias voltage. Specifically, bias voltage VB was set to provide transistors with a drain-source bias voltage close to 1 V, as stated on page 4 from line 152 to 156, in which a sentence on safety has
been included in the revised manuscript.
3. The purpose of this circuit is automotive applications. The automotive industry is one of the most demanding industries and the circuits used in this field must show a high degree of stability and safe operation. How can your proposal meet the safe operation requirements of this industry and overcome the concern associated with the correct and stable biasing?
• Thanks for your comment. Reliability tests have been carried out and the results have been shown in the new Fig. 9(a).
4. How process variations can affect your circuit performance and the risk associated with correct circuit biasing.
• According to the above-mentioned reliability tests, there are no specific risks, as explained in the comment to Fig. 9(a).

Please check the attached file for the replies.

Reviewer 3 Report

The authors presented a power amplifier fabricated using CMOS technology. The amplifier shows pretty high output power and efficiency. This work is sound, and presented in a clear way. I recommend acceptance.

Author Response

The authors presented a power amplifier fabricated using CMOS technology. The amplifier shows pretty high output power and efficiency. This work is sound, and presented in a clear way. I recommend acceptance.

  • Thanks for your comment.

Reviewer 4 Report

It would be useful if the authors can add some discussion on the application of the proposed two-path PA that they have fabricated. 

Author Response

It would be useful if the authors can add some discussion on the application of the proposed two-path PA that they have fabricated.

  • As suggested, some more discussion on the application of the proposed PA has been included on page 1 from line 31 to 35.

Please check the attached file for the replies.

Reviewer 5 Report

In this paper, a 77GHz two-path power amplifier is proposed to overcome the distance and power limitation for automotive radar applications. The power units use a common source driver stage and stacked cascode power stage to achieve efficiency improvement and output power enhancement. The paper is well written and organized. Circuit description is well explained and further verified by experiment. 

I suggest the authors include the measurement of temperature variation in Figures 7 and 8. 

Author Response

In this paper, a 77GHz two-path power amplifier is proposed to overcome the distance and power limitation for automotive radar applications. The power units use a common source driver stage and stacked cascode power stage to achieve efficiency improvement and output power enhancement. The paper is well written and organized. Circuit description is well explained and further verified by experiment. I suggest the authors include the measurement of temperature variation in Figures 7 and 8. 

  • Thanks for comment. The measured output power and PAE as a function of  temperature have been shown in Fig. 9(b).

Round 2

Reviewer 2 Report

I have no concerns.

Reviewer 4 Report

The authors have addressed the comments I raised in the previous round of review. I recommend the paper for publication.